# Alcohol and the Brain–Gut Axis: The Involvement of Microglia and Enteric Glia in the Process of Neuro-Enteric Inflammation

**DOI:** 10.3390/cells12202475

**Published:** 2023-10-18

**Authors:** Mohammed A. S. Khan, Sulie L. Chang

**Affiliations:** 1Department of Neurosurgery, Brigham Hospital for Children, Harvard Medical School, Boston, MA 02115, USA; mkhan5@bwh.harvard.edu; 2Institute of NeuroImmune Pharmacology, Seton Hall University, South Orange, NJ 07079, USA; 3Department of Biological Sciences, Seton Hall University, South Orange, NJ 07079, USA

**Keywords:** first brain, gut microbiome, neuroimmune cells, neuro-enteric glial cells, dysbiosis, second brain, alcohol use disorder

## Abstract

Binge or chronic alcohol consumption causes neuroinflammation and leads to alcohol use disorder (AUD). AUD not only affects the central nervous system (CNS) but also leads to pathologies in the peripheral and enteric nervous systems (ENS). Thus, understanding the mechanism of the immune signaling to target the effector molecules in the signaling pathway is necessary to alleviate AUD. Growing evidence shows that excessive alcohol consumption can activate neuroimmune cells, including microglia, and change the status of neurotransmitters, affecting the neuroimmune system. Microglia, like peripheral macrophages, are an integral part of the immune defense and represent the reticuloendothelial system in the CNS. Microglia constantly survey the CNS to scavenge the neuronal debris. These cells also protect parenchymal cells in the brain and spinal cord by repairing nerve circuits to keep the nervous system healthy against infectious and stress-derived agents. In an activated state, they become highly dynamic and mobile and can modulate the levels of neurotransmitters in the CNS. In several ways, microglia, enteric glial cells, and macrophages are similar in terms of causing inflammation. Microglia also express most of the receptors that are constitutively present in macrophages. Several receptors on microglia respond to the inflammatory signals that arise from danger-associated molecular patterns (DAMPs), pathogen-associated molecular patterns (PAMPs), endotoxins (e.g., lipopolysaccharides), and stress-causing molecules (e.g., alcohol). Therefore, this review article presents the latest findings, describing the roles of microglia and enteric glial cells in the brain and gut, respectively, and their association with neurotransmitters, neurotrophic factors, and receptors under the influence of binge and chronic alcohol use, and AUD.

## 1. Introduction

Alcohol is a central nervous system (CNS) depressant [1]. Generally, alcohol is consumed in different types, such as spirits, wine, beer, etc. [2,3]. Excessive alcohol consumption alters the levels of neurotransmitters and destroys brain cells in the CNS, which are primarily responsible for the decline in cognition and memory [4,5,6,7,8,9]. According to the National Institute of Alcohol Abuse and Alcoholism (NIAAA), alcohol use disorder (AUD) is “*a chronic relapsing brain disorder characterized by an impaired ability to stop or control alcohol use despite adverse social, occupational, or health consequences*”. At the 46th Research Society of Alcoholism—which was held from 24–28 June 2023 in Bellevue, USA—Dr. George F. Koob, Director of the NIAAA, mentioned during his opening lecture that acute and chronic alcohol consumption not only affects the brain, by causing cognitive and neurological disorders [10,11], but also affects every organ of the body, leading to arrhythmia, cardiovascular disease, hepatitis, and pancreatitis, and especially gut dysbiosis [12,13,14,15,16,17]. Binge alcohol consumption is characterized by 3–5 drinks in a short time of 2–3 h, whereas chronic alcohol consumption is considered as the daily intake of alcohol for several weeks [18]. Binge drinking is more common in adolescents, especially among college students. This habit of alcohol addiction initially causes light symptoms, such as lightheadedness and loss of body balance; however, it later turns into unfavorable pathological outcomes (e.g., AUD) [19]. Patients with AUD experience serious consequences, like hypertension, digestive problems, stroke, and alcohol liver disease [20,21,22,23]. In some cases, AUD transforms into different types of cancers, which may be fatal. Alcohol inhibits excitatory neurotransmission as well as activates neuroimmune cells, including microglia, that communicate with each other by elevating the levels of proinflammatory mediators and altering the expressions of a variety of their surface receptors [24]. This dysregulation of immune cells may become worse with the practice of binge or chronic alcohol consumption, leading to the onset of AUD. Alcohol specifically targets the mesolimbic pathway, also known as the reward center, in the midbrain, where it inhibits neuronal activity and activates neuroimmune cells, along with the modulation of neurotransmitters and neurotrophic factors [24,25]. The mesolimbic pathway, which includes midbrain areas such as the VTA, amygdala, hippocampus, and nucleus accumbens, is implicated in AUD [26]. The neurons and glial cells, from the different parts of the brain, actively participate in the process of craving, addiction, withdrawal, relapse, and reward of alcohol [27]. For example, a rare neurological complication termed “Marchiafava-Bignami disease” (MBD) seen in patients with AUD is like motor neuron disease (MND) [28,29]. This MBD is caused by demyelination and the necrosis of the carpus collosum due to chronic alcohol consumption [30]. In alcoholic patients, this kind of pathology occurs due to the changes in the morphologies, molecular structures, and functions of the glial cells, such as microglia, astrocytes, and oligodendrocytes. Microglia per se are also implicated in AUD. Alcohol consumption alters the microglial response, causing neuroinflammation and neurotransmitter dysregulation. Astrocytes provide structural and metabolic support to the neurons by clearing the potassium and neurotransmitters, especially glutamate, in AUD [31,32]. The function of the oligodendrocytes is to myelinate axonal segments and propagate fast saltatory impulses. The coordination between oligodendrocytes and microglia is reported elsewhere, in which oligodendrocytes have been shown to be activated by microglia to synthesize the myelin-related proteins [33]. Binge alcohol consumption also causes alcohol liver diseases, such as alcoholic fatty liver disease and alcoholic steatohepatitis, as well as spleen atrophy [23,34]. However, this topic is out of our scope in this review. This review reports the latest findings on the participation of microglia in coordination with neurotransmitters and neurotrophic factors, and the modulation of pro- and anti-inflammatory mediators in the brain. This review also discusses the roles of enteric glial cells, enteric neurotransmitters, and enteric neurotrophic factors during alcohol-induced gut inflammation/dysbiosis, and how the molecular and biochemical changes occurring in the CNS (first brain) can cause the inflammation/dysbiosis of the gut (second brain) through bidirectional communication, via the vagal afferent (sensory) and efferent (motor) nerves, to regulate the AUD-related gut and brain inflammation, encompassing the brain–gut axis.

## 2. Effect of Alcohol on the First Brain (CNS) and Second Brain (Gut)

Alcohol affects differentially on the CNS, depending on the different areas of the brain. Alcohol can be neurotoxic, as it interferes with communication between the neurons and non-neuronal cells [35,36]. The damaged nerve cells release neurotoxins and excitotoxins, which can alter behavioral attitudes and impair sensations, judgement, movement, vision, motor coordination, and memory [37,38,39] in the cortex, as well as coordination in the cerebellum, by modulating the biochemical pathways [40,41,42]. These neurotoxic changes influence the transmission of impulses within the neuronal network and alter the synaptic activity, eventually leading to unconsciousness. There are different kinds of distilled and undistilled alcohols, which have been shown to have different alcohol contents, volume by volume. Different alcohols possess different neuroinflammatory effects on the CNS that are not only linked to neurological disorders but also to other peripheral diseases, including cardiovascular and bowel diseases [13,43]. In in vitro physiological studies, different concentrations of ethanol (between 2 and 20 mM (low dose) and between 20 and 50 mM (medium dose)) have been used to study the effects on the neural network, imitating the brain regions in vitro [44]. Specific brain regions, such as the VTA, amygdala, prefrontal cortex, thalamus, hippocampus, etc., have also been shown to be affected by equivalent doses in animal models [45]. Like the CNS, the gut also has a prominent neuronal network and neural plexuses (e.g., the myenteric plexus and meissner plexus) and neurotransmitters (e.g., serotonin) [45], which is why it is referred to as a second brain in this review article. During the binge or chronic consumption of alcohol, the CNS responds first—prior to the gut—through the immune system (Figure 1).

### 2.1. Alcohol-Induced Neurotoxicity in Microglia

Microglia constitute the CNS defense system and play multidimensional neuroprotective and neurodestructive roles in AUD (Figure 2). Microglia not only act as housekeepers for maintaining the homeostasis of the brain, but they also clear debris, attenuate neuroinflammation, strip inhibitory synapses, and promote neurogenesis [46]. However, in the hyperactivation state, they become destructive in response to overwhelming inflammatory stimuli during a variety of pathological conditions of neurological diseases [47]. The loss of microglia may lead to the dysfunction of intact neurons, resulting in cognitive deficits. The chronic consumption of alcohol shows that microglia switch themselves to change the expression patterns of genes that are associated with the neuroimmune function [48]. Chronic alcohol consumption impacts more on adults than adolescents, especially elderly people, because adolescents have high rates of recovery from addiction due to their developmental period compared to adults. It is noteworthy that adolescents require greater amounts of alcohol to reach the same levels of intoxication as adults [49]. Marshall et al. demonstrated that binge-like ethanol exposure reduces the number of microglia in the hippocampus and perirhinal and entorhinal cortices of both adolescent and adult rats [50].

### 2.2. Proinflammatory Mediators in Activation of Microglia

Excessive alcohol consumption impairs the neuronal cell function, leading to acute or chronic neuroinflammation, which gradually induces the loss of neurons through programmed cell death (apoptosis), necrosis, and other forms of cell death. In alcoholic patients, it has been shown that binge alcohol consumption produces DAMPs, such as histones, nucleosomes, DNA, mitochondrial DNA, HMGB1, etc., and increases the systemic levels of cytokines [51,52]. Interestingly, alcohol neurotoxicity only kills neurons and not microglia and astrocytes. During neuroinflammation, microglia come first to repair and rescue dying neurons, followed by astrocytes. However, in cases of the exacerbation of alcohol-induced neuroinflammation, these cells become hyperactivated and produce several pathologic sequelae in the CNS. Microglia survive the alcohol insult and switch to the inflammatory phenotype (M1 type) or to an effector cell type. The microglial response to alcohol exposure produces proinflammatory mediators, such as chemokine CC motif ligand 2 (CCL2) [53]. One study showed that a single binge of alcohol prepped microglia for activation, and a double binge activated them to release tumor necrosis factor (TNFα) [50,54]. Another study reported that the systemic inflammation biomarker highly sensitive C-reactive protein (hs-CRP) increased only in men with binge alcohol drinking but not in the women cohort [55,56,57]. It has also been demonstrated that microglial cells attain a distinct morphological status in response to a variety of stimuli. For example, a single stimulus (alcohol) or multiple stimuli (alcohol, endotoxins, and a cocktail of proinflammatory molecules) can activate microglia and release various kinds of cytokines/chemokines, including chemokine C-C ligand 2 (CCL2) [53,58,59]. However, the secretion of CCL2 or other proinflammatory cytokines in the basal or lower levels may not activate microglia but initially prime them to react to secondary stimuli when they encounter higher levels of cytokines in the inflammatory milieu. This cumulative response of microglia actuates them to repair neural circuits or trigger neuroinflammation and the degeneration of nerve endings, causing chronical neuronal loss.

### 2.3. Expressions and Activation of Receptors in Microglia

The regulation of the alcohol-drinking behavior in microglia is mediated by several ionotropic and metabotropic receptors, which are expressed on the microglial surface. The specific deletion of Cannabinoid Receptor 2 (CB2) receptors in microglia and dopamine neurons has been shown to induce neuroinflammation, alter locomotor activity, and induce behavioral changes via alcohol [60]. JWH-133 (dimethylbutyl-deoxy-delta-8-THC), a selective agonist of CB2, has been shown to reduce the cytokine levels, such as CINC-1, -2α/β, and MIP-3α, in parenchymal cells and change microglia towards the non-inflammatory phenotype; however, it could not reverse or ameliorate brain damage [61]. Microglia express several receptors depending on the proinflammatory or anti-inflammatory environment prevailing within the vicinity. A common microglial marker, the ionized calcium-binding adapter molecule 1 (Iba-1), has been shown to be increased after binge ethanol exposure [62,63]. The significance of Toll-like receptors is emerging with regard to alcohol-induced neuroinflammation. Toll-like receptors, such as TLR-1, -2, -yes4, and -9, express under alcohol-induced inflammatory conditions. The postmortem brain tissues of the hippocampal region in alcoholic patients show that TLR-7 mediates neurotoxicity by releasing let-7b macrovesicles [64]. The overexpression of TLR-4 and nod-like receptors (NLRs) also damages the cerebral cortex and hippocampus through the elevation of proinflammatory mediators in response to alcohol drinking, and these neuroinflammatory effects are abolished by the elimination of TLR-4 and NLRs [65,66,67,68,69,70,71]. There are other receptors in the TLR family that have not been investigated in their full capacity in response to binge drinking. In preclinical studies, the ionotropic (e.g., N-methyl-D-asparate (NMDA) and α-amino-3-hydroxy-5-methyl-4-isoxazolepropionic acid (AMPA)) and metabotropic (e.g., GluR1-1, -2, -3, -5, and -7) receptors are associated with the progression of AUD, in which these receptors are targeted as pharmacotherapies, using their agonists and antagonists [72,73,74]. Alcohol shows different effects on purinergic receptors. The P2X4 receptor (P2X4R) is a sensitive purinergic receptor, which is stimulated at the nanomolar concentration of ATP. For example, P2X4R mediates microglial migration, whereas P2X7R increases IL-1β release and pore formation in BV2 microglia [75,76,77]. Gofman et al. demonstrate that P2X4R in microglia responds to binge alcohol that becomes motile after the binding of norepinephrine to the adrenergic receptors [78,79,80]. Microglia express mu (μ) and kappa (κ) but not delta (δ) opioid receptors. The conditional knockout of the μ opioid receptor in microglia decreases the naloxone-induced withdrawal symptoms in female mice [81,82,83]. Other receptors of the cholinergic system (e.g., alpha7 acetylcholine receptors) may also be responsible for the intrinsic changes that occur due to the alcohol effects in microglia [84,85].

### 2.4. Effect of Alcohol on Phenotypic Change in Microglia

As a normal phenotype, microglia (guardian cells) in the CNS play an essential role in performing phagocytic activity. Microglia are the main drivers of neuroinflammation and mediate the onset and progression of AUD. These cells act as first responders to protect the brain from neuroinflammation [86,87]. Microglia sense the changes in the CNS and then immediately switch their phenotype to either proinflammatory M1 or anti-inflammatory M2, or mixed phenotypes [88,89]. The phenotypic switch of microglia in response to inflammatory responses is implicated in various neurological disorders, including AUD. In a polarized form, microglia are proven to be reparative or detrimental, depending on the homeostatic and pathological status of the CNS in response to the consumption of alcohol (e.g., binge or chronic) [89,90,91,92]. Naltrexone is an opioid receptor antagonist that is used as a therapeutic agent against opioid and alcohol abuse. Using the BV-2 microglial cell line, it has been shown that a low dose of Naltrexone can transform microglia from the M1 to the M2 phenotype with signatures of iNOS^high^CD206^low^ and iNOS^low^CD206^high^, respectively [93]. Intracellular changes, like the production of ROS and NOS and the phenotypic-specific expressions of cytokines/chemokines on the gene and protein levels, including surface receptors, are dysregulated in microglia in response to alcohol-induced neuroinflammation. These biochemical derailments affect neurogenesis and neural communication. The function and phenotypes of microglia also vary according to the acute and chronic states of inflammation, the duration of the inflammatory insult, as well as the stage of disease development that activate different signaling pathways. In preclinical animal models and AUD patients, in vivo imaging analyses revealed that microglia are activated with acute as well as chronic alcohol consumption [94,95]. Examination of the transcriptome of rat primary microglial cells exposed to an alcohol and inflammatory cytokine cocktail (TNF-α, IL-1β, and INF-γ) identified 312 and 3000 differentially expressed mRNA transcripts in KEGG pathways and showed an increase in nitrite production. This pathology further increases when the alcohol is combined with the cocktail, regulating the expressions of mRNA transcripts, such as C1qa, b, and c, C3, and C3aR1, of the phagocytosis process [96]. Like macrophages, prior reports of microglial characterization suggested that alcohol-induced neuroinflammation is caused by the microglial phenotype M1 (proinflammatory). When the alcohol effects are blocked, microglia tend to become M2 (anti-inflammatory) [48,54,88,89,93]. However, recent literature unveils that there are more phenotypes of pro- and anti-inflammatory microglia that are driven by different activation stages [97]. It seems that microglia do not change their phenotype to M1 or M2 until the migration of macrophages, which are followed by neutrophils during neuroinflammation. For example, CD11 and CD45 are constitutively expressed by microglia, and these fully activated cells have been shown to increase in the hippocampus and entorhinal cortex, consisting of the CD11^+^CD45^high^ signature, and in rats given alcohol. The same study also analyzed MHC-II, CD32, and CD86 on CD11^+^CD45^low^ and CD11^+^CD45^high^ cells and demonstrated that the majority of CD11^+^CD45^low^ microglial cells are negative for MHC-II, CD86, and CD32 but become positive on CD11^+^CD45^high^ cells after binge alcohol exposure [54,98]. Zareen et al. demonstrated that, during dynamically occurring neurological processes, microglia alter themselves to different morphologies, such as swollen somata, and scattered, round, ameboid, and retracted processes (deramification) in the inflammatory milieu. It is also reported that microglia with diverse populations express MHC-II/OX6 and Iba1 markers. Interestingly, the OX6-positive microglia do not appear until 7 days [97]. In the four-day binge alcohol paradigm, Marshall et al. showed that OX45 is increased, but not OX6 and ED-1 until day 7. They proclaim that OX6 and ED-1 are not expressed due to the partial activation of microglia, which demonstrate the reparative phenotype rather than the destructive phenotype [89].

### 2.5. Impact of Alcohol on Neurotransmitters in the First Brain (CNS)

Alcohol induces a state of euphoria in the reward center of the brain. The frontal lobe of the brain is susceptible to alcohol-related damage. The euphoric effects of alcohol influence different regions of the brain through the mesolimbic system, which is also called the reward center. This mesolimbic circuit or pathway exists in the midbrain (ventral tegmental area (VTA)), where it regulates the autonomic and endocrine functions and transports neurotransmitters, including dopamine, through the dopaminergic neurons in response to binge alcohol consumption. These changes happen through the coordination among interneurons, which originate from the VTA, to other structures of the brain, such as the nucleus accumbens and prefrontal cortex, to sensitize reward-related motivations or addiction [99,100]. For instance, AUD produces psychoactive and addictive behavior in the mesolimbic system. In the VTA, alcohol stimulates GABAergic neurotransmission and modulates the expressions and activities of acetylcholine receptors, including the nicotinic alpha7 acetylcholine receptor. The activation of this mesolimbic system releases dopamine and opiates from the dopaminergic nerve terminal, while alcohol elicits inhibitory effects on glutamate (excitatory neurotransmitter) to act on the neurons present in the nucleus accumbens [101,102,103,104,105]. The amygdala and hippocampus in the temporal lobe play an auxiliary role. During reward, several neurotransmitters are released, including endorphins, endogenous opiates (e.g., morphine-like neurotransmitters), serotonin, GABA, glutamate, and dopamine. The dopamine neurotransmitter is closely associated with reward as well as addiction and reinforcement [106,107]. On the one hand, the release of dopamine from the reward center (i.e., the striatum) acts as a feel-good neurotransmitter. On the other hand, it is responsible for the induction of mechanisms that cause addiction and reinforcement. The neurotransmitters in microglia are also regulated by purinergic receptors with regard to the heavy drinking of alcohol [77]. A study in rats showed that the long-time use of alcohol activates neural sensitization in the mesolimbic circuit that may be responsible for the intense craving and addictive behavior of binge drinking [108,109]. This emphasizes that the feelings of reward or pleasure and repeated behaviors are regulated by the striatum, as it is the center of the reward system in the mesolimbic pathway.

### 2.6. Influence of Alcohol on Neurotrophic Factors in the First Brain

Neurotrophic factors are necessary for the cell proliferation, sustenance, and maintenance of mature and developing neuronal cells in the CNS. In AUD, the neurotrophic factors, such as brain-derived neurotrophic factor (BDNF), glial-derived neurotrophic factor (GDNF), ciliary neurotrophic factor (CTNF), and nerve growth factor (NGF), play essential roles in the regulation of biochemical and metabolic processes via different kinds of receptors in microglia. The release of BDNF has been shown to be elevated after traumatic brain injury to regenerate the damaged brain tissue [110]. In a preclinical study of alcohol-dependent patients, Heberlein et al. showed that BDNF and GDNF modulate addictive behavior after alcohol withdrawal. In this study, the GDNF serum levels were found to be significantly decreased in patients with alcohol dependence, while a decrease in the BDNF levels is associated with alcohol withdrawal [111,112]. In another study, it was shown that alcohol-related changes in the BDNF levels are related to the macrostructure and the processes of polymorphism and neuroplasticity in the brain, whereas increases in the levels of this neurotrophic factor are correlated with cognitive impairment, anxiety, and depression [113,114,115]. Neurotrophic factors regulate the corelease of neurotransmitters and dictate the cholinergic and adrenergic properties of the neuronal cells. For instance, BDNF induces the release of acetylcholine along with norepinephrine in noradrenergic cells. On the contrary, CNTF increases cholinergic and decreases noradrenergic components with affecting excitatory neurotransmission. BDNF and CNTF modulate acetylcholine and norepinephrine via independent signaling pathways to regulate neurotransmission [116,117]. Ceci et al. have discussed the role of NGF with regard to alcohol dependence and its abstinence. With consideration of the available data, they propose that the levels of NGF in the plasma are increased to neutralize the toxic alcohol effects; however, its levels are decreased following alcohol withdrawal. These neurochemical changes are attributed to the epigenetic changes in the methylation of CpG sites in the gene promoter with a decrease in mRNA expression, downregulating the NGF gene during alcohol abstinence or withdrawal [102].

## 3. Alcohol-Induced Inflammation in the Second Brain (Gut)

The gut or gastrointestinal (GI) tract comprises the following parts of the body that are involved in food intake and the discharge of excreta: the mouth, esophagus, stomach, intestine, and rectum. The GI tract accommodates several bacterial species and other varieties of microbes, collectively called “gut microbiota”. The episodic drinking of alcohol can inflame the gut by altering the balance of microbiota and leads to several gut-related problems, such as gastritis, acid reflux, and diarrhea [118]. Acid reflux is caused by the combination of gut inflammation and the dysfunction of the esophageal sphincter muscle, which may, in later stages, lead to anemia and ulcer as well as Barrett’s esophagus or esophageal cancer [119,120]. The gut harbors microbiota, which include several species of bacteria, fungi, and other microbes. Binge alcohol drinking perturbs the normal ratio between harmful and beneficial commensal microbiota by creating an imbalance within the niche of the gut [121,122]. This deviation in the microbial population decreases the absorption of food and the digestion of sugars in the gut and elevates the levels of bile in the liver [123,124,125]. The increase in the pathogenic microbes causes a leaky gut and disrupts the tight junction in the intestinal epithelium (barrier) that allows intestinal flora and toxins to escape into the blood stream, causing diarrhea and systemic inflammation [121,126,127,128]. During the commensal imbalance, the disruption in the digestion of sugars and the shift in the microbial population promote the overgrowth of the non-bacterial species, such as yeast (e.g., *Saccharomyces* and *Candida* sps.) [129]. These fungal species actively participate in the fermentation of the undigested sugars and increase gas production, which causes a bloated and unhealthy gut.

### 3.1. Influence of Alcohol on Neurotransmitters in the Second Brain

Neurotransmitters carry messages among the interneurons within the brain. For instance, the axon terminal of a pre-synaptic neuron releases neurotransmitter-containing vesicles into the synaptic cleft that transmit electric nerve impulses among interneurons and are extended to the muscle in the periphery [130,131]. Interestingly, specialized ascending and descending neurons, collectively known as enteric neurons, are present in the gut [132]. Along with these neurons, enteric glial cells are also inhabitants of the gut. They are one of the prime cell types of the neural crest lineage that perform a similar role in the gut, providing sustenance to the gut, as microglia do in the brain. Acute exposure to alcohol releases excitatory entero-neurotransmitters (e.g., ATP and GABA), which activate P2X2/3 and GABA receptors in excitatory ascending neurons [133,134]. There are several microbes in the gut that specifically synthesize a variety of entero-neurotransmitters using their precursors. Most of the neurotransmitters that are present in the brain also exist in the gut [135]. For example, Chen et al. showed that glutamate is synthesized from their precursors (i.e., acetate by the bacterial species *Lactobacillus plantarum*, *Bacteroides vulgatus*, and *Campylobacter jejuni*) by enteroendocrine cells in the gut, whereas GABA is synthesized from the same precursors as glutamate but with the involvement of different bacteria, such as *Bifodobacterium*, *Bacteroides*, *Parabacteroides*, and *Eubacterium*. The myenteric neurons and mucosal endocrine-like cells are also involved in this synthesis. For acetylcholine synthesis, choline is utilized as a substrate by *Lactobacillus plantarum*, *Bacillus acetylcholini*, *Bacillus subtilis*, *Escherichia coli*, and *Staphylococcus aureus*, and by myenteric neurons. These bacterial species produce 33% of the acetylcholine in the gut. The neurotransmitter serotonin is mainly (95%) produced in the gut and transported to the brain through the blood circulation, which requires 5-hydroxytryptophan (5-HTP) and tryptophan as precursors. These precursors are exploited by *Staphylococcus* and *Clostridia* sps. and produced by enterochromaffin cells. Dopamine, which plays an essential role in reward and addiction, is also synthesized in the gut, utilizing precursors such as amino acid tyrosine and L-3,4-dihydroxyphenylalanine (L-DOPA) by *Staphylococcus* sps. Additionally, a few more neurotransmitters (e.g., norepinephrine, tyramine, tryptamine, and phenylethylamine) are also produced in the gut, using tyrosine, tryptophan, and phenylalanine amino acids, respectively, as their base substances [136]. Thus, enteric neurons and glia, along with the release of neurotransmitters and neurotrophic factors, have the capacity to operate the neural network in the gut as a second brain.

### 3.2. Contribution of Enteric Glial Cells in Alcohol-Induced Gut Inflammation

Like microglia and macrophages in the CNS and PNS, respectively, enteric glial cells are also an essential part of the innate immune system in the enteric nervous system (ENS) and have connections to the neural crest of the gut. These cells resemble Schwann cells at the transcriptional level and demonstrate a similar phenotype in response to various stimuli. It has been shown that the abrogation of glial Gja1 in mouse colon disrupts the intestinal motility, secretomotor reflexes, and neuromuscular functions. Enteric glial cells express markers such as Sox10, Sox2, S100 (a calcium-binding protein), and proteolipid (myelin-associated protein) [137,138,139,140]. These cells are distributed in the walls of mucosa. However, most of the enteric glial cells are present in the enteric ganglia and myenteric plexuses of the ENS. The survival and function of enteric neurons, which lie in the walls of the GI tract (e.g., stomach, intestines, etc.), depend on the performance of enteric glial cells, as their intrusions clean up the neurotoxicity and provide neuroactive substances, sustaining a healthy environment for neurons and their axon terminals [141]. Overall, enteric glial cells protect the parenchymal cells, including intestinal epithelial and enteric neuronal cells, and coordinate with enteric macrophages against the inflammation, cell injury, and infection in inflammatory bowel diseases (e.g., Crohn’s disease and ulcerative colitis) after AUD [43]. However, the role of enteric glial cells in the gut under the influence of alcohol use disorder has not been investigated to a large extent, opening a new avenue for alcohol research.

### 3.3. Influence of Alcohol on Neurotrophic Factors in the Second Brain

Apart from the brain, neurotrophic factors are also released in the gut that take part in cell proliferation, differentiation, and migration, as well as cell survival after exposure to alcohol. Alterations in the levels of neurotrophic (neuro-enteric) factors in the gut have been shown to contribute to bowel diseases [142,143,144,145] in response to binge drinking. These alcohol-induced neuro-enteric factors also modulate the sensation and motility in the gut and are involved in the epithelial barrier function, producing neuroprotection and neuroplasticity to the entire ENS. The ENS is an intricate, large portion of the peripheral nervous system. It is a sort of extension of the CNS and regulates the GI microenvironment independently with very little interference from the CNS. Although it is considered that the enteric-neurotrophic factors are released in the gut only during the early developmental and postnatal period and the maturing of the ENS, emerging data support that the enteric-neurotrophic factors and their receptors are present in adulthood as well as in gut pathophysiology [146]. For instance, GDNF is expressed in the mesoderm and anterior neuroectoderm of the embryonic gut that contribute to the activation of enteric neural crest-derived cells [147,148]. In the gut, these enteric-neurotrophic factors are produced by the microbiota through different mechanisms and pathways, including the kynurenine pathway. Zhu et al. have correlated the chronic alcohol-induced alteration of BDNF and imbalance in the gut microbial population with the neuropsychic behavior [149]. A study shows that the exogenous administration of CNTF decreases the consumption of food by acting on neurons through the promotion of leptin signaling in the hypothalamus [150]. Further research needs to be conducted to know exactly the expression and function of enteric-neurotrophic factors in the gut with binge drinking. Understanding the role of neuro-enteric factors in alcohol dependency, and in withdrawal and relapse after binge drinking, might pave the way towards the development of therapeutic measures, using neuro-enteric factors as potential drugs against AUD and binge drinking-associated gut dysbiosis.

### 3.4. Role of Microbiota in Brain–Gut Axis

During the maintenance of molecular or cellular homeostasis in the brain and gut, a complex network of neurons, nerves, neurotransmitters, and receptors are involved in bidirectional communication via vagal nerve fibers (efferent and afferent), which connect the central and enteric nervous systems (Figure 1). The efferent vagal nerve fibers that originate from the 10th cranial nerve descend to transfer cholinergic anti-inflammatory signals to the gut by releasing neurotransmitters, such as acetylcholine. This neurotransmitter stimulates the alpha7 acetylcholine receptor in macrophages and enteric glial cells, modulating the adverse actions of the gut microbiota/microbiome and alleviating intestinal permeability [151,152]. In contrast to the presence of microbiota/microbiome in the gut, the existence of microbiota in the brain is a new concept, which is vaguely described, and it is speculated that, potentially, resident microbes may exist in the brain. This assumption is made on the basis of the identification of microbial sequences or epitopes both in non-pathological and pathological human brain samples [153]. Apparently, it may be possible that, like gut microbiota, the brain may also harbor microbes as a source to catabolize neurotransmitters enzymatically from the precursors. Many bioactive neurotransmitters, such as acetylcholine, gamma-aminobutyric acid (GABA), serotonin, dopamine, and norepinephrine, are released by neurons, and some of them are produced by microbiota [154]. For example, the neurotransmitters that are produced by microbes are acetylcholine from *Lactobacillus plantarum* and *Bacillus acetylcholini*, dopamine and noradrenaline from *Bacillus* sps., *Escherichia coli*, *Proteus vulgaris*, and *Serratia marcescens*, and GABA from *Bifidobacterium* sps. and *Lactobacillus* sps. Most of the serotonin is produced mainly by *Lactobacillus* sps., *Lactococcus lactis* subsp. *cremoris*, *Escherichia coli*, *Hafnia alvei*, *Morganella morganii*, and *Streptococcus thermophilus* in the gut (see also Section 3.1). These microbes utilize the aromatic amino acids, such as phenylalanine, tryptophan, and tyrosine, as the precursors to produce the neurotransmitters norepinephrine, dopamine, and serotonin, respectively [136,154,155]. The microbiota, in their niche, interact with neurotransmitters either directly, released from efferent vagal fibers, or indirectly, from other sources, to alter the brain activity as well as the gut motility and intestinal permeability. A more comprehensive understanding of the neural activities, network of brain circuits, and interactions of microbiota with neurotransmitters/neuromodulators is an exciting avenue to advance the research domain of the brain–gut axis.

## 4. Alcohol-Induced Modulation of Brain–Liver–Gut Axis

The long-term misuse of alcohol does not only impact the brain and gut but also affects other organs, such as the liver, spleen, pancreas, kidneys, etc. In the liver, heavy alcohol drinking causes several alcohol-related liver diseases by virtue of the fact that this drug-metabolizing organ consists of alcohol-breaking machinery, including alcohol dehydrogenase (ADH), aldehyde dehydrogenase (ALDH2), cytochrome P450 2E1 (CYP2E1), and catalase [156]. These enzymes are integral to decreasing the alcohol content mostly in the liver and blood. Also, these enzyme activities are detected in the brain and gut [157,158]. The dysfunction of these enzymes leads to the generation of toxins as byproducts of alcohol, reactive oxygen species, and fat buildup in the liver, as well as in the brain and gut [159].

The liver acts as a center piece and plays a mediatory role in connecting the first brain (CNS) and the second brain (gut) through the bidirectional vagal nerve, which is modulated by the heavy consumption of alcohol. For instance, in the brain, CYP2E1-mediated alcohol metabolism potentiates binge alcohol-induced neuronal apoptosis. The increased activation of microglia in the brain is due to the rapid increase in the levels of inflammatory cytokines (e.g., TNFα) and reactive oxygen species (ROS) and the activities of transcriptional factors (e.g., NF-κB) [160]. The persistence of these danger signals originating from microglia excite vagal efferent (motor) fibers to extend inflammatory signals through systemic circulation in a cyclic manner to the other two organs within the brain–liver–gut axis [158,160]. Alcoholic patients with cognitive decline have been associated with liver dysfunction (e.g., steatohepatitis) and gut dysbiosis (e.g., gastritis). Recently, another protein, proprotein convertase subtilisin/kexin type 9 (PCSK9), which regulates the plasma low-density lipoprotein cholesterol, has also been implicated in AUD associated with brain and liver inflammation. Along these lines, studies have also shown the increased protein expression and altered methylation of PCSK9 in the cerebrospinal fluid and liver, and that the inhibition of this protein reduced hepatic inflammation and steatosis [161,162]. The identification of the role of PCSK9 may lead to new avenues to understand the molecular mechanisms occurring not only in the brain–gut axis but also in the brain–liver–gut axis.

## 5. Effects of Alcohol on Innate and Adaptive Immune Responses

Immune cells such as lymphocytes, macrophages, and neutrophils are critical mediators of innate and adaptive immunity. These cells are localized in different tissues as specialized cells, for example, as microglia and enteric glial cells in the brain and gut, respectively. These tissue-specific resident cells manifest distinct and unique characteristics and play important roles in the host immune defense system in response to alcohol [163,164]. During alcohol-induced neuroinflammation, microglia become activated and contribute to the modulation of a number of cytokines, including TNFα, which act as messengers to trigger a downstream cascade of signaling molecules. Subsequently, these inflammatory signals influence other immune cell types and their subtypes that generate innate immune responses. Similarly, enteric glial cells or enterocytes also become activated when they sense a perturbation or shift in the colonies of gut microbiota. Alcohol has been shown to increase species of pathogenic bacteria that tend to release endotoxins, such as lipopolysaccharides, which disrupt integral transmembrane proteins (e.g., occludin and claudin) to form a leaky gut and enter into the blood stream, not only leading to inflammation in the brain and gut but also causing systemic inflammation, endotoxemia, and multiple-organ damage [164]. The accumulation of acetaldehyde in mouse gut has also been shown to disrupt the epithelial tight junction due to the significant increase in the intracellular Ca^2+^ levels and ionic currents through the transient receptor vanilloid (TRPV)-6 Ca^2+^-permeable channel in Caco-2 cells. The genetic deletion of TRPV6 in mice prevented alcohol-induced intestinal barrier dysfunction, suggesting the mediatory role of TRPV6 in systemic inflammation and endotoxemia [165].

## 6. Synopsis and Future Directions

There is strong emerging evidence that different areas of the brain, such as the amygdala, hippocampus, and nucleus accumbens in the mesolimbic pathway in the midbrain (VTA), are vulnerable and have prominent cellular, biochemical, or molecular repercussions related to binge alcohol drinking. These areas in the VTA communicate with interneurons. Microglia play an important role and interfere with neuronal activities in these areas to rescue the nerve cells, which are strained by exposure to alcohol. These glial cells also modulate cytokines (e.g., TNFα, CCL2, etc.) by phagocytosing the neurotoxic substances and dead cells resulting from alcohol effects. In a recovery mode, they promote cell proliferation and differentiation by maintaining the levels of neurotrophic factors (BDNF, CNTF, etc.) and neurotransmitters (acetylcholine, serotonin, etc.) in the brain and gut. During the above processes, microglia act uniquely in different stages of neuroinflammation. Therefore, the determination of the microglial phenotype in the brain and the role of enteric glial cells (relative of microglia) in the gut including macrophages, and their morphologies and functions in the alcohol modulation of the brain–gut axis are important to understand in the development of AUD. At the same time, it is also important to note the notion that the brain is the first organ that primarily reacts to alcohol abuse/misuse, subsequently followed by the secondary response from the gut. Thus, the understanding of the molecular and cellular mechanisms and biochemical and metabolic processes, including their signaling pathways, independently in the brain–gut axis might help us discover therapeutics against AUD. A more comprehensive understanding of the neuroimmune activities, network of neural circuits in the first brain (brain), and interactions of neurotransmitters/neuromodulators with microbiota/microbiome in the second brain (gut), especially involving the vagal nerve, is needed because this is an exciting avenue to advance the research domain of the brain–gut axis.

## Figures and Tables

**Figure 1 cells-12-02475-f001:**
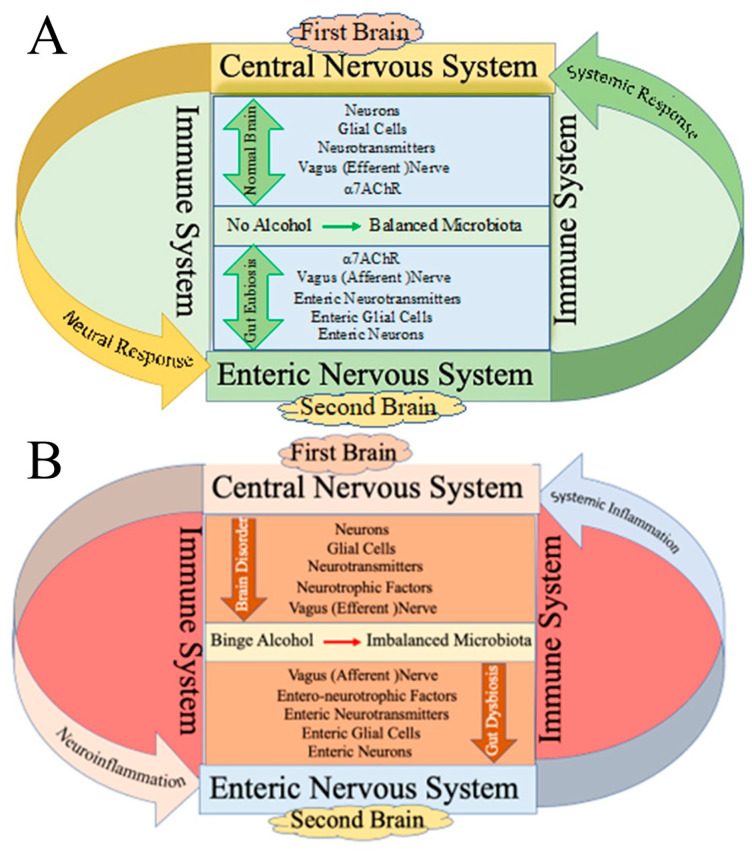
Schematic diagram of connection (bidirectional communication) of first brain (central nervous system (CNS)) and second brain (enteric nervous system ENS)) through the immune system (IMS). (**A**) Bidirectional communication between normal brain and gut in which neurons, glial cells, neurotransmitters, neurotrophic factors, the vagal nerve, and receptors are in homeostatic conditions via the immune system. (**B**) Binge alcohol disrupts the homeostatic conditions causing inflammation in the CNS, ENS, and IMS.

**Figure 2 cells-12-02475-f002:**
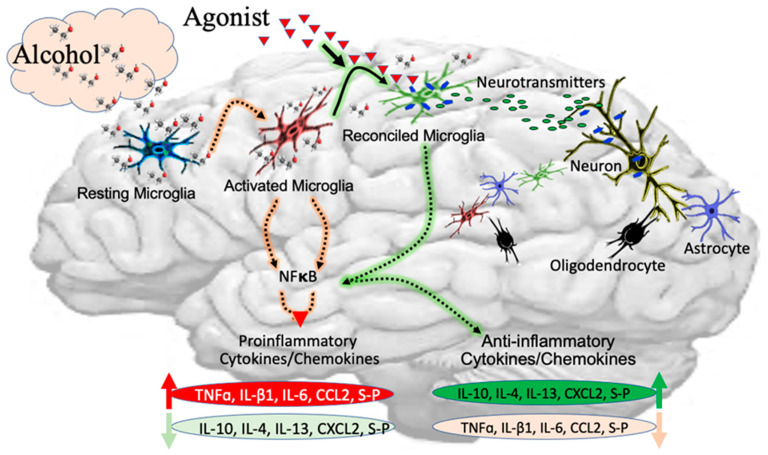
Alcohol-induced neurotoxicity in microglia. Alcohol transforms resting microglia to the activated stage, which induces transcription factors (NFκB and TFEB), leading to an increase in the levels of proinflammatory cytokines and a decrease in anti-inflammatory cytokines. The intervention with anti-inflammatory drugs reconciles the alcohol-induced neuroinflammation, decreases the proinflammatory cytokines, and elevates anti-inflammatory cytokines, regaining the brain homeostasis.

## Data Availability

Not applicable.

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
