# Peer review of "Alcohol and the Brain–Gut Axis: The Involvement of Microglia and Enteric Glia in the Process of Neuro-Enteric Inflammation"

_cells, 2023, doi:10.3390/cells12202475_

Round 1

Reviewer 1 Report

Review article entitled "Alcohol and Brain-Gut Axis: Involvement of Microglia and Enteric Glia in the Process of Neuro-Enteric Inflammation '' is well presented and will move forward the gut brain axis research field. I have few suggestions for the benefit of readers.

Liver plays an important role in liver metabolism, it would be nice if the author can include liver in this review.  1. Author needs to include recent findings in the field (Gut-liver-brain axis). 2. Host-immune responses (innate/adaptive) need to be included in the review article. 3. Impact of endotoxemia and systemic inflammation need to be included in the revised version.  4. Please include the impact of gut microorganism (bacteria, viruces, fungi etc.) on the brain-gut axis. 5. Discuss about future research directions.   

Moderate editing required

Author Response

We appreciate your time to review our manuscript (Cells ID: 2534642) with the title “Alcohol and Brain-Gut Axis: Involvement of Microglia and Enteric Glia in the Process of Neuro-Enteric Inflammation”. Your comments and suggestions were carefully considered. We have revised the manuscript in accordance with the critiques. Below are our point-by-point responses to the  comments as well as the revisions that we made to the manuscript.

Comment #1: Author needs to include recent findings in the field (Gut-liver-brain axis).
Response: We appreciate your suggestion. The recent findings in the field have been reviewed and synopsized under a new Heading (IV. Alcohol-induced inflammation in other organs Modulation of Brain-Liver-Gut Axis). This addition is noted in blue below as well as in the revised manuscript (page 10, lines 439-467).

IV. Alcohol-induced Modulation of Brain-Liver-Gut Axis

For instance, in the brain, CYP2E1-mediated alcohol metabolism potentiates binge alcohol included neuronal apoptosis. Increased activation of microglia in the brain due to the and rapid increase in the levels of inflammatory cytokines (e.g., TNFα), reactive oxygen species (ROS) and activities of transcriptional factors (e.g., NF-κB) [160]. Persistent of these danger signals originating from microglia excite vagal efferent (motor) fibers to extend inflammatory signals through systemic circulation in a cyclic manner to other two organs within the Brain-Liver-Gut Axis [158, 160]. Alcoholic patients with cognitive decline have been associated with liver dysfunction (e.g., steatohepatitis) and gut dysbiosis (e.g., gastritis). Another protein, proprotein convertase subtilisin/kexin type 9 (PCSK9), which regulates the plasma low-density lipoprotein cholesterol. Recently, this protein is also implicated in AUD associated with brain and liver inflammation. Along these lines, studies also showed an increased protein expression and altered methylation of PCSK9 in the cerebrospinal fluid and liver, and inhibition of this protein reduced hepatic inflammation and steatosis [161, 162]. The identification of the role of PCSK9 may lead to new avenues to understand the molecular mechanisms occurring in Brain-Gut Axis but also in Brain-Liver-Gut Axis.  

Comment #2: Host-immune responses (innate/adaptive) need to be included in the review article.
Response: We appreciate your input. A paragraph on innate and adaptive responses with a heading of V. Effects of Alcohol on Innate and Adaptive Immune Responses has been added. This addition is noted in blue below as well as in the revised manuscript (page 10-11, lines 468-490).

V. Effects of Alcohol on Innate and Adaptive Immune Responses

Immune cells such as lymphocytes, macrophages and neutrophils are critical mediators of innate and adaptive immunity. These cells are localaized in different tissue as specialized cells as, for example, microglia and entretic glial cells in the brain and gut, respectively. These tissue -specific residents cells manifest distinct and unique characteristics and play important roles in host immune defense system in response to alcohol [163, 164]. During the alcohol-induced neuroinflammation, microglia get activated and contribute to the modulation of a number of cytokines including TNFα, which act as a messenger to trigger downstream cascade of signaling molecules. Subsequently, these inflammatory signals influence other immune cells types and their subtypes that generate innate of immune responses. Similarly, enteric glial cells or enterocytes also get activated when they sense the perturbation or shift in the colonies of gut microbiota. Alcohol has been shown to increase species of pathogenic bacteria that tend to release to endotoxins such as lipopolysaccharide, which disrupts integral transmembrane proteins (e.g., occludin and claudins) to form a leaky gut and enter into the blood stream, leading to inflammation not only in the brain and gut but also causing systemic inflammation and endotoxemia and multiple organ damage [164]. Accumulation of acetyldehyde in mice gut has also shown to disrupt epithelial tight junction due to the significant increase in intracellular Ca2+ levels and ionic currents through the transient receptor vanilliod (TRPV)-6, Ca2+ permeable channel, in Caco-2 cells. Genetic deletion of TRPV6 in mice prevented alcohol-induced intestinal barrier dysfunction, suggesting the mediatory role of TRPV6 in systemic inflammation and endotoxemia [165].

Comment #3: Impact of endotoxemia and systemic inflammation need to be included in the revised version. 
Response: We thank you for your suggestion. The information about endotoxemia and systemic inflammation has been included in the paragraph V. Effects of Alcohol on innate and adaptive immune responses. This addition is noted in blue below as well as in the revised manuscript (page 11, lines 480-490).   

Alcohol has been shown to increase species of pathogenic bacteria that tend to release to endotoxins such as lipopolysaccharide, which disrupts integral transmembrane proteins (e.g., occludin and claudins) to form a leaky gut and enter into the blood stream, leading to inflammation not only in the brain and gut but also causing systemic inflammation and endotoxemia and multiple organ damage [164]. Accumulation of Acetaldehyde in mice gut has also shown to disrupt epithelial tight junction due to the significant increase in intracellular Ca2+ levels and ionic currents through the transient receptor vanilloid (TRPV)-6, Ca2+ permeable channel, in Caco-2 cells. Genetic deletion of TRPV6 in mice prevented alcohol-induced intestinal barrier dysfunction, suggesting the mediatory role of TRPV6 in systemic inflammation and endotoxemia [165].

Comment #4: Please include the impact of gut microorganism (bacteria, viruces, fungi etc.) on the brain-gut axis.
Response: We thank you for your suggestion. Effect of gut microbiota has been described in the subsection (iv) Role of microbiota in Brain-Gut Axis. This addition is noted in blue below as well as in the revised manuscript (pages 9-10, line 407-438).

(iv) Role of microbiota in Brain-Gut Axis

During the maintenance of molecular or cellular homeostasis in the brain and gut, a complex network of neurons, nerves, neurotransmitters and receptors are involved in a bidirectional communication via vagal nerve fibers (efferent and afferent), which connect central and enteric nervous systems (Fig. 1). The efferent vagal nerve fibers that originate from 10th cranial nerve descend to transfer cholinergic anti-inflammatory signals to the gut by releasing neurotransmitter such as acetylcholine. This neurotransmitter stimulates alpha7 acetylcholine receptor in macrophages and enteric glial cells, modulating the adverse actions of gut microbiota/microbiome and alleviating intestinal permeability [151, 152]. In contrast to the presence of microbiota/microbiome in the gut, the existence of microbiota in the brain is a new concept, which is vaguely described and speculated that potentially the tiny microbes may exist in the brain. This assumption is made on the basis of identification of microbial sequences or epitopes both in non-pathological and pathological human brain samples [153]. Apparently, it may be possible that, like gut microbiota, the brain may also harbor microbes as a source to catabolize neurotransmitters enzymatically from the precursors. Many bioactive neurotransmitters such as acetylcholine, gamma-aminobutyric acid (GABA) serotinin, dopamine, norepinephrine are released by neurons and some of the them are produced by microbiota [154]. For example, the neurotransmitters that are produced by microbes are; acetylcholine from Lactobacillus plantarum and Bacillus acetylcholini, dopamine and noradreniline from Bacillus sps., Escherichia coli, Proteus vulgaris and Serratia marcescens, GABA from Bifidobacterium sps., and Lactobacillus sps. Most of the serotonin is produced mainly by Lactobacillus sps., Lactococcus lactis subsp. cremoris, Escherichia coli, Hafnia alvei, Morganella morganii and Streptococcus thermophilus in the gut (also see section III-i). These microbes utilize the aromatic amino acids such as phenylalanine, tryptophan and tyrosine, as the precursors to produce neurotransmitters norepinephrine, dopamine and serotinin, respectively [136, 154, 155]. The microbiota, in their niche, interact with neurotransmitters either directly, released from efferent vagal fibers, or indirectly, from other sources, to alter the brain activity as well as gut motility and intestinal permeability. A more comprehensive understanding of the neural activities, network of brain circuits and the interactions of microbiota with neurotransmitters/ neuromodulators is an exciting avenue to advance the domain of Brain-Gut Axis.

Comment #5: Discuss about future research directions. 
Response: We thank you for your suggestion. The future directions have been included in this review article as a part of synopsis and future direction, in the section VI. Synopsis and future directions. This addition is noted in blue below as well as in the revised manuscript (page 11, line 511-515).

A more comprehensive understanding of the neuroimmune activities, network of neural circuits in first brain (brain), and the interactions of neurotransmitters/neuromodulators with microbiota/microbiome in the second brain (gut), especially involving vagal nerve, is needed because this is an exciting avenue to advance the domain of Brain-Gut Axis.

Reviewer 2 Report

This is an interesting review regarding the effects of ethanol on inflammation, glial cell phenotype, and the brain-gut axis. This is a relatively understudied area of research, the review is timely, and will serve as a good resource for the field.

The authors should address the following concerns:

1)      The manuscript requires significant editing to correct grammatical errors prior to publication.

2)      In the Abstract, the following sentence needs to be modified and clarified- “These glial cells derived from mesoderm and present in the resting stage, alone with astrocytes and oligodendrocytes, acting as scavengers that dynamically survey the CNS”. The meaning of this sentence is unclear. It appears to appears to attempt to link several unrelated topics. Also, very little is said about astrocytes and oligodendrocytes in the manuscript, so inclusion in the Abstract seems unwarranted.

3)      Line 37- “different areas in the brain in the CNS”- this needs to be clarified.

4)      Line 41- Refers to recent RSA meeting. If June 24-28 is included, the year should also be included.

5)      Fix font around line 197 to match the remainder of the manuscript.

Needs significant editing.

Author Response

We appreciate your time to review our manuscript (Cells ID: 2534642) with the title “Alcohol and Brain-Gut Axis: Involvement of Microglia and Enteric Glia in the Process of Neuro-Enteric Inflammation”. Your comments and suggestions were carefully considered. We have revised the manuscript in accordance with the critiques. Below are our point-by-point responses to the  comments as well as the revisions that we made to the manuscript, which have been highlighted in blue font.

Comment #1: The authors should address the following concerns:
1)      The manuscript requires significant editing to correct grammatical errors prior to publication.
2)      In the Abstract, the following sentence needs to be modified and clarified-
 “These glial cells derived from mesoderm and present in the resting stage, alone with astrocytes and oligodendrocytes, acting as scavengers that dynamically survey the CNS”. The meaning of this sentence is unclear.
Response: We have carefully edited our revised manuscript to be sure that the grammatical details are correct. The sentence you have noted above has been rewritten. The revision is noted in blue below as well as in the revised manuscript (page1, lines 16-18)

Microglia, like peripheral macrophages, are an integral part of immune defense and represent the reticuloendothelial system in the CNS. Microglia constantly survey the CNS to scavenge the neuronal debris.

Comment #2: It appears to appears to attempt to link several unrelated topics.
Response: We thank you for your comments. The irrelevant information has been removed.
Our review manuscript has said very little about astrocytes and oligodendrocytes, so inclusion of these two cells in the Abstract does not seem to be warranted. Instead, astrocytes and oligodendrocytes have been omitted from the abstract.

Comment #3: Line 37- “different areas in the brain in the CNS”- this needs to be clarified.
Response: The above phrase has been rewritten. The revision is noted in blue below as well as in the revised manuscript (page 1, lines 34-36).

Excessive alcohol consumption alters the levels of neurotransmitters and destroys neuronal cells in the CNS that are primarily responsible for the decline in cognition and memory.

Comment #4: Line 41- Refers to recent RSA meeting. If June 24-28 is included, the year should also be included.
Response: We thank for your attention on the specific detail. This addition is noted in blue below as well as in the revised manuscript (page 1 line 40)

In 46th Research Society of Alcoholism - which was held from June 24-28, 2023 in Bellevue, USA

Comment #5: Fix font around line 197 to match the remainder of the manuscript.
Response: Again, we thank you for your attention on the specific detail. The font has been checked and assured to be consistent throughout the revised manuscript (see below page 5 line 195-198).

Microglia express mu (µ), and kappa (κ) but not delta (δ) opioid receptors. The conditional knockout of µ opioid receptor in microglia decreases the naloxone-induced withdrawal symptoms in female mice.